# Applied Assessment Method for Varus Thrust during Walking in Patients with Knee Osteoarthritis Using Acceleration Data Measured by an Inertial Measurement Unit

**DOI:** 10.3390/s22176460

**Published:** 2022-08-27

**Authors:** Shogo Misu, So Tanaka, Kohei Ishihara, Tsuyoshi Asai, Tomohiko Nishigami

**Affiliations:** 1Department of Physical Therapy, Faculty of Nursing and Rehabilitation, Konan Women’s University, 6-2-13 Morikita-machi, Higashinada-ku, Kobe 658-0001, Hyogo, Japan; 2Department of Rehabilitation, Fukuoka Orthopedic Hospital, 2-10-50 Yanagochi, Minami-ku, Fukuoka 815-0063, Fukuoka, Japan; 3Department of Orthopedic Surgery, Fukuoka Orthopedic Hospital, 2-10-50 Yanagochi, Minami-ku, Fukuoka 815-0063, Fukuoka, Japan; 4Faculty of Rehabilitation, Kansai Medical University, 18-89 Uyamahigashicho, Hirakata 573-1136, Osaka, Japan; 5Department of Physical Therapy, Faculty of Health and Welfare, Prefectural University of Hiroshima, 1-1 Gakuen-tyou, Mihara 723-0053, Hiroshima, Japan

**Keywords:** gait, thrust, knee, osteoarthritis, inertial measurement unit, acceleration

## Abstract

We developed a novel quantitative method to assess varus thrust during walking using acceleration data obtained from an inertial measurement unit (IMU). This study aimed to examine the reliability of the developed index and to evaluate its ability to distinguish patients with knee osteoarthritis (OA) with varus thrust from healthy adults. Overall, 16 patients with knee OA and 16 healthy adults walked on a treadmill with IMUs attached to the tibial tuberosity and lateral femoral condyle. As an index of varus thrust, we used the root mean square (RMS) of acceleration in the mediolateral direction. This value was adjusted by dividing it by swing speed while walking (adjusted RMS, A-RMS) because the RMS of the acceleration was strongly coupled with the speed of motion. The intraclass correlation coefficients of A-RMS of the tibia and femur were 0.85 and 0.73, respectively. Significant differences were observed in the A-RMSs of the tibia and femur, with large effect sizes between the patients with knee OA and healthy adults (Cohen’s d: 1.23 and 0.97, respectively). Our results indicate that A-RMS has good test–retest reproducibility and can differentiate patients with varus thrust from healthy adults.

## 1. Introduction

Knee osteoarthritis (OA) is a chronic degenerative disease that develops and progresses because of aging and other related factors. The symptoms of knee OA include pain, stiffness, reduced joint motion, and abnormal gait patterns [1,2]. Long-term consequences include reduced physical activity, deconditioning, impaired sleep, fatigue, depression, and disability [3]. Because the rate of symptom progression and disease severity varies among individuals, objective assessments of the symptoms are important not only to determine the extent of progression but also to evaluate the effectiveness of treatment.

Varus thrust is an abnormal gait pattern frequently (16–46%) observed in patients with medial knee OA [4,5,6,7,8,9,10,11]. This abnormal gait pattern is defined as an excessive “bowing-out” knee motion in the frontal plane during ambulation as the limb accepts weight with a return toward a more neutral alignment in late stance and swing [4]. Several reports have shown that the varus thrust during walking is associated with knee adduction moments [4,12,13,14]. Mechanical stress on the knee joint is considered one of the factors affecting disease progression. Hence, the varus thrust of the knee joint has been reported to be associated with radiographic severity and disease progression [4,7,15]. Moreover, varus thrust has been reported to be associated with pain severity and functional ability [8,9,10]. In a prospective cohort study of patients with knee OA, varus thrust was related to increased incidence of pain as well as clinically important worsening pain [11].

A few studies have attempted to quantitatively assess varus thrust during walking using some equipment. Many of these studies were based on observations by experts, such as physical therapists, and generally categorized gait into the presence or absence of varus thrust [4,5,6,7,8,9,10,11]. A three-dimensional motion analysis system is a tool used to objectively quantify varus thrust [6,12,16,17,18]. A study reported that patients with OA had a more varus knee angle during early and mid-stance measured by the system versus controls [12]. However, this assessment method has several limitations and is not widely used clinically. Motion analysis systems are expensive, require a long time to prepare for measurements, and involve notable efforts to collect a large amount of data for testing epidemiological hypotheses.

An inertial measurement unit (IMU) is an alternative tool for assessing varus thrust during motion. It is comparatively cheap and requires little preparation for measurement. Acceleration data, which can directly measure the impact transmitted to the location of the sensor, could be good indicators of varus thrust. This is attributable to the fact that the knee is subjected to greater forces and moves in the lateral direction during the beginning of the stance phase in patients with thrust, which would result in greater acceleration. Although several researchers have quantified varus thrust using acceleration data from IMUs [14,19,20], the validity of their methods appeared to be insufficient. As the magnitude of acceleration during walking in each part of the body is considered to have a strong relationship with the speed of motion of the measured location [21,22,23], we believed that the index representing varus thrust should be adjusted on the basis of the angular velocity. However, to the best of our knowledge, no studies have considered such adjustments in their reports of the assessment of varus thrust. Additionally, these studies did not compare thrust measurements between patients with knee OA and healthy adults, and the cutoff value to determine a normal gait was unclear.

In this study, we developed a new method to assess varus thrust during walking motion using acceleration data obtained from an IMU attached to the tibia and femur to adjust for the angular velocity obtained from the same IMU. This study aimed to examine the reliability and validity of the developed method by evaluating the test–retest reproducibility of the derived index of varus thrust and whether this index could distinguish patients with knee OA with varus thrust from healthy adults. We hypothesized that the new adjusted method would have a more discriminative ability than the unadjusted method.

## 2. Materials and Methods

### 2.1. Subjects

This study included 16 patients diagnosed with knee OA (10 women) and 16 healthy adults (10 women). The subjects in the knee OA group were recruited from an orthopedic hospital and were scheduled for tibial osteotomy. The inclusion criteria for the knee OA group were as follows: (1) symptomatic knee OA, diagnosed according to the clinical or radiographic classification criteria of the American College of Rheumatology; (2) Kellgren and Lawrence (K-L) classification of II or more; and (3) age 45–80 years. The healthy group consisted of employees aged ≥ 45 years who worked at the institution that recruited the knee OA group. The study excluded those who had difficulty walking unaided or had other musculoskeletal or neurological diseases that clearly affected walking, such as hip OA, spinal canal stenosis, stroke, or Parkinson’s disease. All patients in the knee OA group were confirmed to have varus thrust in the affected knee while walking based on the observation of the therapist in charge. The reliability of this assessment method has been previously reported [10]. The demographic data and basic characteristics of the knee OA group were extracted from their medical records. The Japanese version of the Knee Injury and Osteoarthritis Outcome Score (KOOS) was used to evaluate the disease status [24]. These assessments and gait analyses were performed as preoperative assessments a week before the surgery.

### 2.2. Gait Assessment

The subjects walked on a treadmill at a comfortable speed chosen by the participant. Before the assessment, subjects practiced sufficiently to familiarize themselves with the walking conditions. Elastic belts were used to secure a total of four IMUs (MVP-RF8; MicroStone, Nagano, Japan) to the bilateral tibial tuberosities and lateral femoral condyles (Figure 1). The IMU was equipped with a three-axis acceleration sensor (range: ±60 m/s^2^) and a three-axis gyroscope (range: ±2000 deg/s), and the obtained data were wirelessly transmitted to a personal computer via Bluetooth. Acceleration in the anteroposterior (AP), mediolateral (ML), and vertical (VT) directions and angular velocity in the sagittal, frontal, and horizontal planes were measured at a sampling rate of 200 Hz. To test the reproducibility of the measurements, the subjects performed the same assessments under the same conditions within a week.

### 2.3. Data Analysis

The data obtained from the IMUs were analyzed using commercially available software (MATLAB, Release 2019a; MathWorks Japan, Tokyo, Japan) to calculate the index of varus thrust during walking motion. Data for ten strides during stable walking on the affected side in the knee OA group and on the left side in the healthy group were used for the analysis. Angular velocity data were sent through a low-pass filter with a cutoff frequency of 20 Hz. At first, heel contact and toe-off were identified from the VT and AP acceleration data of the tibia, respectively, according to previous studies [25,26]. Thereafter, the acceleration data in the ML directions of the femur or tibia in the first halves of the stance phase were extracted (Figure 2) because the knee joint thrust occurs during this period [12]. The root mean square (RMS) of the extracted acceleration data (RMS of the acceleration data of the halves of the stance phase, stRMS) in the ML direction was calculated as an index of varus thrust before adjustment (Equation (1)). We used the RMS as an index because we believed that the magnitude of the entire waveform series during the halves of the stance phase repeatedly demonstrated large positive and negative values immediately after initial contact. Next, we extracted the angular velocity data in three directions during the swing phase and calculated the RMS (RMS of the angular velocity data of the swing phase, v-swRMS) of each plane data in the femur and the tibia (the Equation (2)). As a final index of varus thrust, the adjusted RMS (A-RMS) of the femur or the tibia was calculated by dividing stRMS in the ML direction by the mean of v-swRMSs of the femur or the tibia, respectively, in the three planes (the Equation (3)) because RMS of the acceleration was considered strongly coupled with the swing speed of the lower extremity.
(1)$$\mathrm{stRMS}=\sqrt{\frac{1}{N_{st}}\overset{N_{st}}{\underset{i=1}{\sum }}{\left(a_{i}\right)}^{2}}$$
(2)$$v-\mathrm{swRMS}=\sqrt{\frac{1}{N_{sw}}\overset{N_{sw}}{\underset{i=1}{\sum }}{\left(v_{i}\right)}^{2}}$$
(3)$$A-\mathrm{RMS}=\frac{\mathrm{stRMS}}{\mathrm{mean}\;\mathrm{of}\;v\_\mathrm{swRMSs}\;\mathrm{in}\;3\;\mathrm{planes}}$$
where *N_st_* is the number of datapoints in the halves of the stance phase, and ***a****_i_* is the *i*th acceleration datapoint. *N_sw_* is the number of datapoints in the swing phase, and ***v****_i_* is the *i*th angular velocity datapoint. The three planes include the sagittal, frontal, and coronal planes.

### 2.4. Statistical Analysis

The demographic characteristics of the knee OA and healthy groups were compared using unpaired *t*-tests or χ^2^ tests. To examine the association between stRMS in the ML direction and mean v-swRMS, we calculated Pearson’s correlation coefficients. To assess the reproducibility of A-RMS, the intraclass correlation coefficient (ICC_1,1_) was calculated. Unpaired *t*-tests were used to examine whether there were differences in stRMS in the ML direction or in A-RMS in the ML direction in the tibia or femur between the knee OA and healthy groups. The effect size of each comparison was calculated using Cohen’s *d*. If a significant difference was noted between the groups for A-RMS during univariate analysis, multiple regression analysis was used to ascertain the difference after adjustments for age and body mass index (BMI). Additionally, receiver operating characteristic (ROC) curves were created, and the area under the curve (AUC) was calculated using A-RMS as the dependent variable and the knee OA group as the independent variable. The Youden index was used to calculate the cutoff value, sensitivity, and specificity for classifying the two groups. The level of statistical significance was set at <5%. All statistical analyses were performed using JMP version 14.3.0 (SAS Institute Inc., Cary, NC, USA).

## 3. Results

The characteristics of the subjects are presented in Table 1. The mean age of the knee OA group (62.8 ± 6.5 years) was higher than that of the healthy group (53.4 ± 4.6 years, *p* < 0.001). The mean BMI was also higher in the knee OA group than in the control group (26.6 ± 4.6 vs. 22.7 ± 2.2 kg/m^2^, *p* < 0.001). The number of subjects per K-L classification in the knee OA group was four and twelve for classes II and III, respectively; no subjects were K-L class IV. The mean sub-scores of KOOS in the knee OA group ranged from 33.2 for quality of life to 65.8 for activities of daily living.

The association between stRMS in the ML direction and the mean v-swRMS in the three planes are shown using scatter plots (Figure 3). Pearson’s correlation coefficients were 0.49 (*p* < 0.01) and 0.30 (*p* = 0.09) for the tibia and femur, respectively.

The mean values and standard deviation of stRMS and A-RMS in the ML direction of the tibia and femur for two measurements to assess reproducibility are presented in Table 2. ICC_1,1_ of A-RMS in the ML direction of the tibia and femur was 0.85 and 0.73, respectively. The mean stRMS and A-RMS in the knee OA and healthy groups are presented in Table 3. Although no significant differences were observed in stRMS between the knee OA and healthy groups, there were significant differences in the A-RMSs of the tibia and femur, with large effect sizes between the two groups (tibia, 1.23; femur, 0.97). Multiple regression analysis revealed that knee OA was significantly related to A-RMS in the ML direction of the tibia after adjustment for age (standardized *β* = 0.63, *p* = 0.005). Another model adjusting for BMI also showed a significant relationship (standardized *β* = 0.37, *p* = 0.03). However, the model including both age and BMI as covariates did not show a significant association between knee OA and A-RMS (standardized *β* = 0.41, *p* = 0.08).

ROC curves were created to assess the discriminative ability of A-RMS (Figure 4). The AUC of A-RMS in the ML direction of the tibia was 0.81, with 75.0% sensitivity and 81.3% specificity (cutoff: 0.027 m/s/deg). The AUC of A-RMS in the ML direction of the femur was 0.75, with 68.8% sensitivity and 81.3% specificity (cutoff: 0.038 m/s/deg).

## 4. Discussion

We developed a new method to assess varus thrust during walking using ML acceleration data measured by IMUs attached to the tibia or femur. We proposed an index, A-RMS, which is stRMS in the ML direction adjusted by dividing it by the mean value of v-swRMS in the three planes. At first, we examined whether stRMS in the ML direction was correlated with the mean v-swRMS. Thereafter, we examined the test–retest reproducibility of A-RMS and showed that the index had moderate to good reproducibility. Finally, we evaluated the validity of the index. The A-RMSs of the tibia and femur were significantly larger in those with varus thrust than in those without varus thrust, with a large effect size, although the unadjusted RMSs were not significantly different between the groups. Additionally, A-RMS had good ability to discriminate between the presence and absence of varus thrust. Particularly, the AUC of A-RMS of the tibia demonstrated high accuracy (AUC = 0.81). Therefore, we consider that A-RMS of the tibia is a useful index for quantitatively assessing varus thrust during walking motion.

A moderate correlation was observed between stRMS in each position and the mean v-swRMS in the three measured planes. A faster speed of motion before the foot contacts the ground during walking (swing speed) is expected to produce a greater impact to the lower extremity, and consequently, a greater degree of acceleration. Therefore, this was a reasonable result. In past studies, the RMS of acceleration in the trunk was shown to be related to gait speed and was used after adjusting for gait speed as an index of trunk movement during walking [21,22,23]. The method used in this study to assess varus thrust was similar to these; the RMS of acceleration in the ML direction of the femur or tibia was adjusted by dividing it by the angular velocity of the same position. We used the mean v-swRMS in the three planes, which was considered to represent the speed of the lower limb more accurately.

The results of the evaluation of reliability indicated that A-RMS in the ML direction of the tibia and femur had high reproducibility (ICC_1,1_ of 0.85 and 0.73, respectively). The ICC values were equivalent to those obtained for the RMS of acceleration of the lower trunk in healthy older adults [27]. Additionally, the RMS of acceleration in the ML direction of the shank or thigh showed similar ICC values in patients with knee OA for the test–retest evaluation (shank 0.92 and thigh 0.70) [28]. Considering that ICC values between 0.5 and 0.75 and between 0.75 and 0.9 indicate moderate and good reproducibility, respectively [29], A-RMS of the tibia and femur had moderate to good reproducibility and was a reliable measure.

To the best of our knowledge, this is the first study to demonstrate that an index of varus thrust calculated from acceleration data (adjusted by dividing it by angular velocity) measured by IMUs attached to the lower extremities could discriminate well between patients with knee OA with varus thrust and healthy adults. However, stRMS, which did not undergo adjustment based on the velocity of the position where the IMU was attached, was not a good discriminator. Several past studies support our results. Yoshimura et al. reported that the peak acceleration of the tibia in the ML direction was significantly greater in knees with ruptured anterior cruciate ligaments and varus thrust than in contralateral normal knees and knees that had undergone anterior cruciate ligament reconstruction [19]. Ishii et al. reported that the peak lateral acceleration of the tibia was higher in knees with severe OA than in knees with early-stage OA [20]. However, these studies did not make adjustments for the speed of the lower extremity or assess thrust by observation. Although a study showed that the first peak of lateral acceleration with adjustment for gait speed was significantly higher in grade 3 and 4 OAs (severe) than in grade 0 OA [30], the study did not assess varus thrust. The results of these studies were similar to ours, but unlike them, we compared the varus thrust measure adjusted for the speed of sensor motion between patients with knee OA with varus thrust and healthy adults without thrust, and our findings demonstrated good discriminative ability. Tsukamoto, et al. demonstrated that the peak knee varus angular velocity measured by an IMU could also distinguish the gait with thrust from that without thrust [31]. The AUC was 0.896 with a sensitivity of 0.957 and a specificity of 0.579, which was comparable to our results. Further investigation is needed to examine which parameter from acceleration or angular velocity can distinguish varus thrust more precisely.

In the clinical setting where rehabilitation is performed, the assessment of varus thrust during walking by observation is often employed to provide disease prognosis or verify the effectiveness of interventions. This study was notably significant in that it showed that the presence or absence of varus thrust could be determined using IMUs, which are easy to use in clinical settings. Additionally, it showed the cutoff value with a high degree of accuracy, which is useful for clinical gait assessment. However, the study did not clarify whether this method could be used to assess the severity of varus thrust. The subjects in this study were scheduled for tibial osteotomy, and their K-L classification was II or III. Thus, almost all subjects demonstrated a moderate degree of varus thrust during walking, but we could not assess the severity of the thrust. Because previous studies have reported that more severe thrust is associated with more severe functional disability and larger knee adduction moments [14,18], assessing the degree of thrust is important. A report indicated that the peak knee varus angular velocity was associated with the severity of trust [31]. Further studies are needed to examine the association between A-RMS and the severity of thrust in patients with knee OA of various severities.

This study had some limitations. First, this was a single-center study with a small sample size, which makes generalizing the results problematic. Particularly, the mean KOOS for the knee OA group was higher than previously reported for patients with knee OA and for those scheduled for high tibial osteotomy, indicating that the knees in our sample were in relatively good condition [24,32]. Second, there were several potential confounders of the relationship between knee thrust and A-RMS. For example, there were significant differences in the mean age and BMI between the knee OA and healthy groups. Age and BMI have been reported to affect acceleration during walking [33,34,35]. Because strict matching based on both parameters could not be implemented for clinical reasons, we performed statistical adjustments. However, the association between the group and A-RMS after adjustment for age and BMI was not significant. This could have been caused by a lack of statistical power owing to the small sample size. Additionally, other potential confounding factors that we did not measure, such as muscle strength or pain, may also affect the results. Future studies should take these limitations into account.

## 5. Conclusions

A-RMS, which is a novel index for assessing varus thrust using acceleration data adjusted by dividing it by angular velocity, showed good test–retest reproducibility. The A-RMSs of the tibia and femur had good discriminative ability for varus thrust; in particular, A-RMS of the tibia demonstrated high accuracy. The index may be a useful index for the quantitative assessment of varus thrust during walking. Further studies involving more subjects with varying severity of thrust are warranted to confirm the usefulness of the A-RMS in clinical settings.

## Figures and Tables

**Figure 1 sensors-22-06460-f001:**
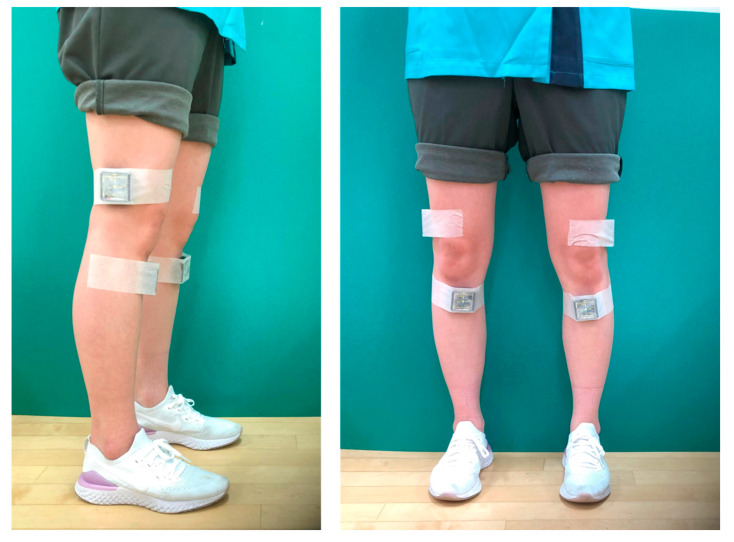
Photograph of the inertial measurement units attached to the tibia and femur.

**Figure 2 sensors-22-06460-f002:**
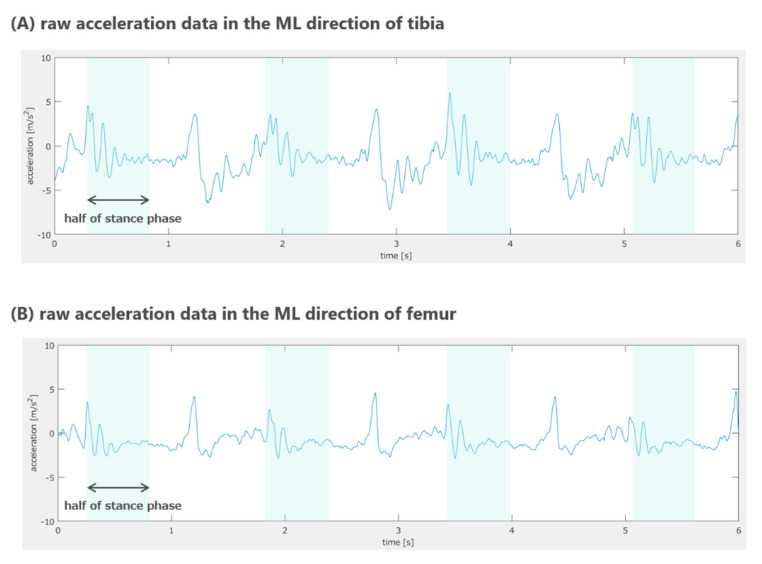
Typical raw acceleration data in the ML direction for the tibia and femur around the knee and the extracted data for the assessment of varus thrust. The upper and lower figures represent the raw acceleration data of the tibia (**A**) and femur (**B**) for several typical gait cycles during the walking state. Data for the sections colored light green were for each of the first halves of the stance phases, and they were extracted to calculate the index of varus thrust. ML, mediolateral.

**Figure 3 sensors-22-06460-f003:**
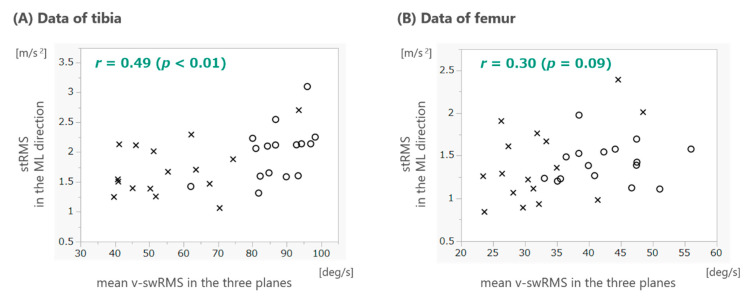
Scatter plots between stRMS in the ML direction and the mean v-swRMS in the three planes. The left and right figures represent the plot of the data for the tibia (**A**) and femur (**B**), respectively. The cross marks indicated the values of patients with knee OA, and the circle marks indicate the values of healthy adults. Pearson’s correlation coefficient, *r*, is shown in the upper left corner of each plot. RMS, root mean square; stRMS, RMS of the acceleration data of half of the stance phase; v-swRMS, RMS of the angular velocity data of the swing phase; ML, mediolateral.

**Figure 4 sensors-22-06460-f004:**
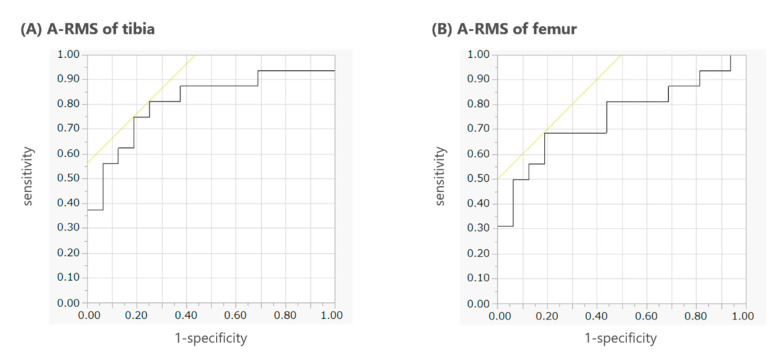
ROC curves of A-RMS for detecting patients with knee OA and varus thrust. The left and right figures represent the ROC curve of A-RMS in the ML direction of the (**A**) tibia (AUC: 0.81) and (**B**) femur (AUC: 0.75). ROC, receiver operating characteristic; A-RMS, adjusted root mean square of the acceleration data of half of the stance phase; OA, osteoarthritis; ML, mediolateral; AUC, area under the curve.

**Table 1 sensors-22-06460-t001:** Comparison between the characteristics of subjects in the knee osteoarthritis and healthy groups.

Variables		Knee OA Group	Healthy Group	*p* Value
		(*n* = 16)	(*n* = 16)	
Age	(years)	62.8 ± 6.5	53.4 ± 4.6	<0.001
Females/males	(n)	10/6	10/6	1.00
Body weight	(kg)	67.7 ± 16.7	60.5 ± 7.2	0.12
Height	(m)	1.59 ± 0.11	1.63 ± 0.08	0.22
Body mass index	(kg/m^2^)	26.6 ± 4.6	22.7 ± 2.2	0.005
Gait speed	(km/h)	1.26 ± 0.35	1.98 ± 0.26	<0.001
Affected side: Left/right	(n)	6/10	—	—
K-L classification (II/III/IV)	(n)	4/12/0	—	—
KOOS score				
Symptom score		63.8 ± 10.8	—	—
Pain score		55.1 ± 16.4	—	—
ADL score		65.8 ± 16.1	—	—
Sports score		34.8 ± 21.8	—	—
QOL score		33.2 ± 16.4	—	—

Values are presented as mean ± standard deviation or percentages. *p* values were calculated using unpaired *t*-tests or χ^2^ tests between the knee OA and healthy groups. OA, Osteoarthritis; K-L classification, Kellgren and Lawrence classification; KOOS, Knee Injury and Osteoarthritis Outcome Score; ADL, activity of daily living; QOL, quality of life.

**Table 2 sensors-22-06460-t002:** Index of varus thrust for two measurements within a week and the intraclass correlation coefficients (*n* = 32).

Variables		Day 1	Day 2	ICC_1,1_
stRMS in the ML direction at tibia	(m/s^2^)	1.86 ± 0.47	1.96 ± 0.54	0.70
stRMS in the ML direction at femur	(m/s^2^)	1.41 ± 0.35	1.45 ± 0.37	0.65
A-RMS in the ML direction at tibia	(m/s/deg)	0.027 ± 0.008	0.029 ± 0.010	0.85
A-RMS in the ML direction at femur	(m/s/deg)	0.039 ± 0.011	0.039 ± 0.012	0.73

Values are presented as mean ± standard deviation. *p* values were calculated using unpaired *t*-tests between the knee OA and healthy groups. ICC, intraclass correlation coefficient; stRMS, root mean square of acceleration data of half of the stance phase; A-RMS, adjusted root mean square of acceleration data of half of the stance phase; ML, mediolateral.

**Table 3 sensors-22-06460-t003:** Index of varus thrust between the knee osteoarthritis and healthy groups.

Variables		Knee OA Group	Healthy Group	Effect Size	*p* Value
		(*n* = 16)	(*n* = 16)	(Cohen’s *d*)	
stRMS in the ML direction at tibia	(m/s^2^)	1.71 ± 0.45	2.00 ± 0.46	0.64	0.08
stRMS in the ML direction at femur	(m/s^2^)	1.40 ± 0.45	1.42 ± 0.23	0.06	0.83
A-RMS in the ML direction at tibia	(m/s/deg)	0.032 ± 0.009	0.023 ± 0.005	1.23	0.002
A-RMS in the ML direction at femur	(m/s/deg)	0.044 ± 0.013	0.034 ± 0.007	0.97	0.010

Values are presented as mean ± standard deviation. *p* values were calculated using unpaired *t*-tests between the knee OA and healthy groups. OA, Osteoarthritis.

## Data Availability

The data presented in this study are only available to members of this project. The data will not be shared with others because consent for data sharing was not obtained from the participants.

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
