# Peer review of "Applied Assessment Method for Varus Thrust during Walking in Patients with Knee Osteoarthritis Using Acceleration Data Measured by an Inertial Measurement Unit"

_sensors, 2022, doi:10.3390/s22176460_

Round 1

Reviewer 1 Report

Dr. Misu and colleagues demonstrate a new method and utility for quantifying varus thrust in patients with knee OA using an inertial measurement unit. This finding is clinically significant and interesting. I am satisfied with the quality of work presented.

There have been several unclear issues, so I have a few questions and suggestions for corrections.

In line123-130, Probably, the adjustment in angular velocity when calculating femoral and tibial A-RMS is performed at the angular velocity of the femur and tibia, respectively. To make the method clearer for general audiences, please describe the point in more detail.

The results of Pearson’s correlation coefficient are written in lines 184-186 and figure 3, but I suggest adding a p-value.

Is it possible to change the shape of the plots in figure 3 for the knee OA group and the healthy group? (For example, the shape of Plots in the OA group is round, and the shape of Plots in the healthy group is a triangle.) The increase in stRMS in the ML direction depending on the angular velocity of the swing phase is better represented. If it is technically difficult, no modification is required.

In lines 306-313, I have a question about the authors' discussion of the results of multiple regression analysis. The author considers to lack of statistical power that the no significant model including both age and BMI as covariates did not show a significant association between knee OA and A-RMS, is this due to sample size? Are there other possible causes?

Author Response

Comments to the Reviewer #1

We would like to thank Reviewer #1 for his or her useful comments, which we have used to revise the manuscript. We believe that our manuscript has been substantially improved as a result of these changes. Detailed point-by-point responses to the reviewer’s comments are shown below. In the revised manuscript, the new or modified text is highlighted.

Reviewer’s comments:

> Dr. Misu and colleagues demonstrate a new method and utility for quantifying varus thrust in patients with knee OA using an inertial measurement unit. This finding is clinically significant and interesting. I am satisfied with the quality of work presented.

Response: Thank you for this positive comment.

> There have been several unclear issues, so I have a few questions and suggestions for corrections.

> In line123-130, Probably, the adjustment in angular velocity when calculating femoral and tibial A-RMS is performed at the angular velocity of the femur and tibia, respectively. To make the method clearer for general audiences, please describe the point in more detail.

Response: Thank you for your comments. The femoral and tibial A-RMS were calculated by adjusting the RMS of the femur and tibia’s angular velocity data, respectively. So, several words were added in the revised manuscript (Pg.4, lines 137–140).

> The results of Pearson’s correlation coefficient are written in lines 184-186 and figure 3, but I suggest adding a p-value.

Response: Thank you for your comment. The p-value was added in the revised manuscript (Pg.5, line 200) and figure 3 (Pg.6).

> Is it possible to change the shape of the plots in figure 3 for the knee OA group and the healthy group? (For example, the shape of Plots in the OA group is round, and the shape of Plots in the healthy group is a triangle.) The increase in stRMS in the ML direction depending on the angular velocity of the swing phase is better represented. If it is technically difficult, no modification is required.

Response: Thank you for your useful comments. The shape of the plots in figure 3 were changed according to your comments (Pg.6) and the explanations about that were added in the figure legends (Pg.6, lines 215–216).

> In lines 306-313, I have a question about the authors' discussion of the results of multiple regression analysis. The author considers to lack of statistical power that the no significant model including both age and BMI as covariates did not show a significant association between knee OA and A-RMS, is this due to sample size? Are there other possible causes?

Response: Thank you for your comments. The lack of statistical power was considered due to the small sample size, because the standardized β after adjusting for both age and BMI was 0.41, which indicated the possibility of moderate association after adjustments, and was equivalent to that after adjusting for BMI only.

The sampling bias and the other potential confounding factors might also affect the results. The healthy control group subjects might have more muscle strength since they were younger than the knee OA group. These differences might affect the results. Furthermore, most knee OA patients were considered to have knee pain during walking, which might result in a gait pattern with weakened impact at initial contact.

Therefore, we considered that further studies were warranted to take into account those limitations. In the revised manuscript, the several sentences were added to explain these (Pg.10, lines 347–349).

Reviewer 2 Report

1.         What mechanism was thought for A-RMS index to distinguish between a healthy person and another that presents knee OA?

Please provide the analysis and the math behind the A-RMS index.

2.         Since you are using RMS signals, important information about waveforms is lost, this limits your A-RMS index to be used in person with no additional condition.

How does the feasibility of you’re A-RMS change when used with persons with Parkinson's or other diseases?

How does speed affect your index?

3.         Please provide a comparison between your proposed index and other technique in the state of the art.

4.         Signals processing looks too simple, Is it necessary the use of Matlab?

5.         What are the units for Fig 3?

             What red line does represent in Fig 3?

Author Response

Comments to the Reviewer #2

We would like to thank Reviewer #2 for his or her useful comments, which we have used to revise the manuscript. We believe that our manuscript has been substantially improved as a result of these changes. Detailed point-by-point responses to the reviewer’s comments are shown below. In the revised manuscript, the new or modified text is highlighted.

Reviewer’s comments:

> 1. What mechanism was thought for A-RMS index to distinguish between a healthy person and another that presents knee OA?

Response: Thank you for your comment. Varus thrust is an abnormal gait pattern defined as an excessive “bowing-out” knee motion in the frontal plane during ambulation as the limb accepts weight with a return toward a more neutral alignment in late stance and swing [reference 4]. This is often seen in patients with knee OA. Thus, the ML direction acceleration of the femur and tibia during the beginning of stance phase is expected to be greater in patients with knee OA than in healthy adults. That is why an A-RMS was applied as the varus thrust index. Acceleration data usefulness to represent varus thrust had already been reported [reference 14, 19, 20] in some studies. Those were described in more detail in the Introduction section (Pg.2, lines 63–77). Additionally, a sentence was added to describe the above (Pg.2, lines 66–68).

> Please provide the analysis and the math behind the A-RMS index.

Response: Thank you for your comment. We added the math equations to calculate the A-RMS in the revised manuscript (Pg.4, lines 143–151).

> 2. Since you are using RMS signals, important information about waveforms is lost, this limits your A-RMS index to be used in person with no additional condition.

How does the feasibility of you’re A-RMS change when used with persons with Parkinson's or other diseases?

How does speed affect your index?

Response: Thank you for your invaluable comments. The waveforms information is lost when RMS is used as you have indicated. For example, the acceleration peak value was not calculated in this study. However, the value of RMS is thought to include some information regarding the peak value given the waveform characteristics. The RMS was used as a varus thrust index because the magnitude of the entire waveform series during the halves of stance phase which repeatedly took large positive and negative values immediately after initial contact (see Figure 2) might indicate the degree of thrust. Thus, the A-RMS is believed to be useful for varus thrust assessment. If additional information is needed, such as the peak value or the other parameters indicating the quality of movement, those can be further analyzed by using the raw data waveforms. In the revised manuscript, a sentence was added to explain why we used RMS as a varus thrust index (Pg.4, lines 132–135).

A-RMS was intended to be used only under normal gait speed condition, and was not intended to be used under other additional conditions. Several previous studies demonstrated that varus thrust assessment only under normal condition was enough to provide disease prognosis [reference 7, 11, 15]. This indicator is not considered to be applicable other than to knee OA, such as Parkinson’s diseases, since varus thrust is a characteristic symptom of patients with knee OA and its appearance in other patients is quite rare. Therefore, the results of this study are considered to be sufficiently applicable to the clinical practices of patients with knee OA.

With regards to the speed, the effect of gait speed is believed to already be taken into consideration since the RMS of acceleration was adjusted by RMS of angular velocity during the swing phase (Pg.4, lines 138–141). Lower extremity velocity during the swing phase is a measure that is closer to walking speed and it was considered that making an adjustment using that was more effective in removing effect of speed on acceleration.

> 3. Please provide a comparison between your proposed index and other technique in the state of the art.

Response: Thank you for your invaluable suggestion. Several sentences in the Discussion section was added based on your comment in the revised manuscript (Pg.9 lines 313–318).

> 4. Signals processing looks too simple, Is it necessary the use of Matlab?

Response: Thank you for your comment. The signal processing in this study was indeed simple, but requires somewhat tedious work in identifying the gait events and calculating the RMS for each gait period. Thus, Matlab was used for data analysis. However, similar analysis can be done using simpler and easily available software, and we hope to implement our thrust index using such software when it is applied clinically in the future. Thank you very much for your invaluable comment.

> 5. What are the units for Fig 3?

       What red line does represent in Fig 3?

Response: Thank you for your comments. The units for figure 3 (Pg.6) were added based on your comments. The red line was deleted in the revised manuscript because linear regression analysis was not performed in this study, while this line in the original manuscript represented the regression line.

Reviewer 3 Report

Below I'm sharing some thoughts which may be of interest to the Authors, but I think this paper has no serious flaws and may be published in its present form.

The Authors state that "to test the reproducibility of the measurements, the subjects performed the same assessments under the same conditions within a week" (l. 108–110), but the results of the repeated experiments are not indicated in the paper and it is unclear how these results were used to assess the reproducibility of the studied method.

In l. 235–236 the Authors claim that they have “confirmed” the correlation between stRMS in the ML direction and mean v-swRMS. However, the results presented in Figure 3 indicate that that correlation is rather weak. Nevertheless, the weakness of that correlation does not seem to undermine the validity of the studied method.

It is unclear what the Authors mean by the expression “preadjusted RMS” in l. 240. Perhaps “unadjusted RMS”?

It is unclear what the Authors mean by the expression “good discriminative ability regardless of varus thrust” in l. 241. I think that the ability to discriminate between the presence and absence of knee osteoarthritis with varus thrust is the only discriminative ability studied in this paper.

It is unclear what the Authors mean by stating that “acceleration data depend on the speed of the measured position” and that this dependence explains the “moderate correlation” between stRMS and v-swRMS (l. 247–248). The fact that acceleration is the first derivative of velocity does not entail their correlation (cf. e.g. sinusoidal oscillation).

The statements provided in Section 5 (e.g. “The A-RMS of the tibia is a useful index for the quantitative assessment of varus thrust”) do not seem to be sufficiently supported by the experimental results presented in this paper, in particular – taking into account certain limitations of the study, described at the end of Section 4. More experiments, involving more subjects, seem necessary to decisively confirm the discriminative ability of the A-RMS indicator and its usefulness for quantifying varus thrust. Perhaps a more restrained conclusion regarding the applicability potential of the A-RMS indicator would be more adequate in Section 5.

The expression “stRMS adjusted by v-swRMS” is consistently used throughout the paper. Perhaps the expression “stRMS divided by v-swRMS” or “stRMS adjusted by dividing it by v-swRMS” would be more precise.

In Table 2 the term femoris is used instead of femur.

The expression “to predict disease prognosis” in l. 286 is unfortunate, because the verbs “to predict” and “to prognose” are almost equivalent; it could be replaced with either “to predict disease development” or “to provide disease prognosis”.

In l. 62, the expression “low cost” is used as an adjective, so there should be a hyphen (“low-cost”); however, “cheap” seems better.

Author Response

Comments to the Reviewer #3

We would like to thank Reviewer #3 for his or her useful comments, which we have used to revise the manuscript. We believe that our manuscript has been substantially improved as a result of these changes. Detailed point-by-point responses to the reviewer’s comments are shown below. In the revised manuscript, the new or modified text is highlighted.

Reviewer’s comments:

> Below I'm sharing some thoughts which may be of interest to the Authors, but I think this paper has no serious flaws and may be published in its present form.

Response: Thank you for this positive comment.

> The Authors state that "to test the reproducibility of the measurements, the subjects performed the same assessments under the same conditions within a week" (l. 108–110), but the results of the repeated experiments are not indicated in the paper and it is unclear how these results were used to assess the reproducibility of the studied method.

Response: Thank you for your comment. The table was added based on the comment to present the mean value and standard deviation, as well as the intraclass correlation coefficient of the repeated experiments’ varus thrust index (Pg. 7, lines 225–227, Table 2).

> In l. 235–236 the Authors claim that they have “confirmed” the correlation between stRMS in the ML direction and mean v-swRMS. However, the results presented in Figure 3 indicate that that correlation is rather weak. Nevertheless, the weakness of that correlation does not seem to undermine the validity of the studied method.

Response: Thank you for your comments. We are in agreement that the correlation was rather weak. We changed the word “confirmed” to “examined” in the revised manuscript (Pg.8, line 268).

> It is unclear what the Authors mean by the expression “preadjusted RMS” in l. 240. Perhaps “unadjusted RMS”?

Response: We apologize for the unclear phrasing. Your point was correct, and the word was replaced in the revised manuscript (Pg.8, line 273). Thank you for pointing this out.

> It is unclear what the Authors mean by the expression “good discriminative ability regardless of varus thrust” in l. 241. I think that the ability to discriminate between the presence and absence of knee osteoarthritis with varus thrust is the only discriminative ability studied in this paper.

Response: We are very sorry that the phrasing error caused you trouble once again. As indicated, the discriminative ability was examined as to whether the varus thrust was present or absent. The sentence was corrected in the revised manuscript (Pg.8, lines 274–275).

> It is unclear what the Authors mean by stating that “acceleration data depend on the speed of the measured position” and that this dependence explains the “moderate correlation” between stRMS and v-swRMS (l. 247–248). The fact that acceleration is the first derivative of velocity does not entail their correlation (cf. e.g. sinusoidal oscillation).

Response: Thank you for your insightful comment, which we fully agree with. The “acceleration data depend on the speed of the measured position” description was unclear and misleading.

Nevertheless, lower limb acceleration has a strong relationship with the speed of motion because the change in speed of motion over a given period of time (such as stride cycle) is greater when a person walks faster, considering the walking motion characteristics. In fact, the RMS of acceleration in the trunk was shown to be strongly related to gait speed in the previous studies [reference 24–26]. The faster the speed of motion before the foot contacts the ground during walking (swing speed), the greater the impact to the lower extremity is expected to be, and consequently, the greater the acceleration. The sentences about the rationale of why acceleration during the halves of stance phase was related to the angular velocity during swing phase in the revised manuscript (Pg.8, lines 280–282).

> The statements provided in Section 5 (e.g. “The A-RMS of the tibia is a useful index for the quantitative assessment of varus thrust”) do not seem to be sufficiently supported by the experimental results presented in this paper, in particular – taking into account certain limitations of the study, described at the end of Section 4. More experiments, involving more subjects, seem necessary to decisively confirm the discriminative ability of the A-RMS indicator and its usefulness for quantifying varus thrust. Perhaps a more restrained conclusion regarding the applicability potential of the A-RMS indicator would be more adequate in Section 5.

Response: Thank you for your comments and we fully agree with that. We changed the description in the Conclusion in the revised manuscript (Pg.10, lines 361–364).

> The expression “stRMS adjusted by v-swRMS” is consistently used throughout the paper. Perhaps the expression “stRMS divided by v-swRMS” or “stRMS adjusted by dividing it by v-swRMS” would be more precise.

Response: Thank you for your comment. The description in the revised manuscript was replaced (Pg.1, line 25; Pg.8, line 267, 287; Pg.9, lines 302; and Pg.10, line 359).

> In Table 2 the term femoris is used instead of femur.

Response: Thank you for your comment. The word in the revised manuscript was changed, but the table number was noted to be changed from 2 to 3 owing to table addition (Pg.7).

> The expression “to predict disease prognosis” in l. 286 is unfortunate, because the verbs “to predict” and “to prognose” are almost equivalent; it could be replaced with either “to predict disease development” or “to provide disease prognosis”.

Response: Thank you for your comment. The word in the revised manuscript was changed based on your comment (Pg.9, line 326).

> In l. 62, the expression “low cost” is used as an adjective, so there should be a hyphen (“low-cost”); however, “cheap” seems better.

Response: Thank you for your comment. The word in the revised manuscript was changed (Pg.2, line 66).

Round 2

Reviewer 2 Report

Thanks I have no more concerns